# Trends in the burden of gastroesophageal reflux disease in China and Global from 1990 to 2021 and predictive analysis for 2040

Zexu Tang[1], Zelin Chen[1], Yongjian Huang[1¤a], Jinwen Fang[2], Xiangyang Yang[2], Chunzhu Li[2], Huiying Zhang[2]*, E. Xie [2¤b]*

**1** Graduate School, Guangzhou University of Traditional Chinese Medicine, Guangzhou, Guangdong, China, **2** Department of the Center for Gastroesophageal Relux Disease, Shantou Hospital of Traditional Chinese Medicine, Shantou, Guangdong, China

¤a Current Address: Department of Academic Research, Shantou Hospital of Traditional Chinese Medicine, Shantou, Guangdong, China.
¤b Current Address: Department of the Center for Gastroesophageal Relux Disease, Shantou Hospital of Traditional Chinese Medicine, Shantou, Guangdong, China.
* 1121532184@qq.com (HZ); 1621981300@qq.com (EX)

## Abstract

### Background

This study analyzed the temporal trends in the age- and gender-specific disease burden of gastroesophageal reflux disease (GERD) in China from 1990 to 2021, comparing its incidence, prevalence, and disability-adjusted life years (DALYs) with global estimates.

### Methods

Data were extracted from the Global Burden of Disease (GBD) database 1990–2021. The Joinpoint regression model was used to quantify trends via annual average percentage change (AAPC) and 95% confidence intervals. Multidimensional comparisons were conducted by age, gender, and time.

### Results

From 1990 to 2021, China exhibited declining trends in age-standardized incidence, prevalence, and DALYs rates for GERD, in contrast to increasing trends globally. Disease burden was higher among females and increased with age, with the middle-aged and elderly experiencing the greatest rise. Projections indicate a continued increase in GERD burden in China by 2040.

**Data availability statement:** All relevant data are within the paper and its Supporting Information files.

**Funding:** The author(s) received no specific funding for this work.

**Competing interests:** NO authors have competing interests Enter: The authors have declared that no competing interests exist.

**Abbreviations:** AAPC, Average annual percentage change; APC, Annual percentage change; ASR, Age-standardized rate; ASDR, Age-standardized disability-adjusted life years rate; ASIR, Age-standardized incidence rate; ASPR, Age-standardized prevalence rate; BAPC, Bayesian age-period-cohort; GERD, Gastroesophageal Reflux Disease; GBD, Global Burden of Disease; CI, Confidence interval; DALYs, Disability-adjusted life years; LES, Lower esophageal sphincter; DisMod-MR 2.1, Disease Modeling Meta-Regression version 2.1; PPIs, Proton Pump Inhibitors.

## Conclusion

Despite recent declines, GERD remains a significant public health challenge in China due to its large population and rapid aging. Targeted prevention and intervention strategies are urgently needed.

## Introduction

GERD is a chronic, relapsing gastrointestinal disorder primarily characterized by heartburn and acid regurgitation [1]. Over the past decades, GERD has emerged as a major contributor to global non-fatal disease burden. Recent updates from large-scale epidemiological studies indicate that the absolute number of individuals affected by GERD continues to rise worldwide, largely driven by population growth and demographic aging, even in regions where age-standardized rates appear stable or declining [2]. This epidemiological transition underscores the importance of evaluating both relative and absolute disease burden when assessing long-term trends.

Despite advances in pharmacological management, approximately 20%–40% of patients experience an inadequate response to standard-dose proton pump inhibitors (PPIs), resulting in refractory GERD (RGERD). This condition not only compromises symptom control and quality of life but is also closely associated with anxiety, depression, and increased healthcare utilization [3–5], Recent observational studies further suggest that persistent symptoms despite treatment contribute substantially to indirect economic costs and productivity losses, highlighting GERD as a condition with significant individual and societal implications [6]. Consequently, effective prevention and population-level management of GERD remain critical public health priorities [7].

Several GBD studies have systematically quantified the incidence, prevalence, and DALYs associated with GERD at global and regional levels. Earlier analyses based on GBD 2017 and GBD 2019 datasets reported a steady increase in global GERD burden, with marked heterogeneity across regions and a strong association with socioeconomic development indicators [8].More recent GBD updates have refined disease modeling methods and expanded data sources, improving the robustness of estimates while confirming that GERD remains one of the leading causes of digestive disease–related disability worldwide. However, these global assessments primarily emphasize cross-national comparisons and regional aggregation, often obscuring country-specific trajectories, particularly in populous and rapidly aging nations such as China.

China represents a unique epidemiological context due to its vast population size, rapid demographic transition, and evolving lifestyle patterns. Although several national and subnational studies have reported increasing GERD prevalence in selected regions or clinical populations [9–11].most of these investigations are limited by short observation periods, restricted geographic coverage, or reliance on self-reported symptoms. Moreover, prior large-scale analyses rarely provide a comprehensive evaluation of long-term temporal trends stratified by age and sex within the Chinese population. Importantly, discrepancies in diagnostic criteria, data sources, and analytical approaches across studies have led to inconsistent estimates, complicating direct comparisons with global GBD findings and limiting their policy relevance.

Recent international research has emphasized that trends in age-standardized rates alone may underestimate the true public health impact of chronic diseases in aging societies [12]. Studies focusing on GERD and other non-fatal digestive disorders have demonstrated that declining or stable standardized rates can coexist with substantial increases in absolute case numbers and DALYs, driven primarily by population aging and growth. This phenomenon highlights the necessity of integrating decomposition analyses and age–period–cohort approaches to disentangle demographic and epidemiological drivers of disease burden—an aspect insufficiently addressed in existing China-focused GERD studies.

A precise and comprehensive understanding of the epidemiology and temporal dynamics of GERD (ICD-11: DA22) is therefore essential for evidence-based health policy formulation. The lack of high-resolution, long-term analyses grounded in standardized global datasets hampers the development of targeted prevention strategies and efficient resource allocation. Without reliable age- and sex-disaggregated evidence, it remains challenging to identify vulnerable subgroups and anticipate future healthcare demands associated with GERD.

To address these gaps, the present study leverages the most recent GBD 2021 dataset to conduct a detailed 32-year (1990–2021) analysis of GERD burden in China, with explicit comparisons to global trends. By applying advanced statistical methods—including joinpoint regression, decomposition analysis, and Bayesian age–period–cohort (BAPC) modeling—we systematically examine age- and gender-specific patterns and project disease burden through 2040. Through direct comparison with prior GBD-based findings and other large-scale GERD studies, this work provides novel insights into China's distinct epidemiological trajectory and contributes robust evidence to inform tailored public health strategies in the context of rapid population aging.

## Materials and methods

### Data sources

The GBD 2021 study represents a comprehensive global observational epidemiological investigation. It integrates a vast array of 100,983 data sources, encompassing vital registration systems, verbal autopsies, censuses, household surveys, disease – specific registries, health – service contact data, and other relevant information.

This study adopted a broad geographic scope, encompassing 204 countries and territories worldwide. For analytical feasibility, these geographic units were stratified into 21 regions and 7 super-regions to facilitate systematic data processing. The research focused on 371 diseases, with core outcomes including the assessment of incidence, prevalence, and disability rates for each condition. In the assessment process, multiple key covariates were incorporated to ensure comprehensiveness, namely age and sex distributions across populations, geographical disparities between regions, and temporal trends over the study period. By integrating these contextual factors, the study enables a detailed analysis of disease patterns under diverse scenarios, thereby providing robust data support and a rigorous analytical foundation for subsequent research endeavors. For China, the data on the GERD burden from 1990 to 2021 were gathered via the Global Health Data Exchange and its associated results tool (https://vizhub.healthdata.org/gbd – results/) [13].

Regarding GBD 2021, relevant GERD data, including incidence, mortality, and disability rates, along with their corresponding age – standardized values, were selected for both China and the global context from 1990 to 2021. These data were stratified by age groups (5–9 years old, with each 5 – year age group up to 95 years old, and those 95 years old and above) and calendar years from 1990 to 2021.

Since the GBD 2021 data are publicly accessible, the institutional ethics committee did not require ethical review for this particular study. Moreover, this study adheres to the guidelines for accurate and transparent reporting of health assessments [14].

### Statistical analysis

The GBD study integrates estimates from a wide range of disease and injury outcome models. Detailed documentation of the key statistical methods, mathematical frameworks, and executable code is publicly accessible via the GHDx website

(https://ghdx.healthdata.org/gbd-2021/code). The primary standardization approaches include the Cause of Death Ensemble Model (CODEm) and the Bayesian meta-regression tool Modeling Meta-Regression version (DisMod-MR), which are employed to estimate prevalence, incidence, mortality, and DALYs [15]. Importantly, uncertainty intervals (UIs) were used to quantify the uncertainty surrounding all estimated outcomes. DisMod-MR generates 500 posterior draws for each estimate, and the 95% UIs were defined as the 2.5th and 97.5th percentiles of these draws. These UIs reflect the combined uncertainty arising from data availability, model specification, and parameter estimation. Comprehensive descriptions of data sources, variable definitions, statistical modeling techniques, and data quality enhancement initiatives have been extensively documented in prior publications [16,17].

To identify years with statistically significant shifts in temporal trends (joinpoints), we utilized Joinpoint Regression software (version 5.1.0.0; National Cancer Institute, USA; https://surveillance.cancer.gov/joinpoint/download). We computed the APC, AAPC, and their corresponding 95% confidence intervals (95% CI). The significance of trend variations across segments was assessed by testing whether the AAPC differed from zero, with a P-value < 0.05 considered statistically significant [18,19].Goodness-of-fit for the Joinpoint model was evaluated using the Bayesian Information Criterion (BIC), and sensitivity analyses were conducted by varying the maximum number of joinpoints to ensure the robustness of the identified trends.

The decomposition method was applied to attribute changes in GERD burden from 1990 to 2021 to three primary factors: population aging, population growth, and epidemiological changes. This approach allowed disentanglement of the overall burden trend into its key drivers, offering clearer insights into how demographic and epidemiological transitions influenced temporal patterns [20,21].

The BAPC model was employed to project future disease burdens. Owing to its ability to comprehensively represent temporal trends and account for complex interactions among age, period, and cohort effects, the BAPC model is well-established in epidemiological research, particularly for age-structured population data [22,23].Model validation included the use of posterior predictive checks and comparison with held-out data to ensure projection reliability. Sensitivity analyses were performed by varying prior distributions and model assumptions to assess the robustness of the 2040 burden forecasts. Utilizing GBD 2021 data and population projections from the Institute for Health Metrics and Evaluation (IHME), this model facilitates detailed and validated predictions of future GERD burden. Both statistical analysis and data visualization were carried out using R (version 4.4.1).

## Result

### Incidence, prevalence, and disability – adjusted life years of GERD in China and Worldwide from 1990 to 2021

**Incidence.** Between 1990 and 2021, the number of GERD cases in China escalated from 20,863,747 (95% CI: 18,034,679–23,688,721) to 32,387,866 (95% CI: 27,851,900–36,711,150), marking a cumulative rise of 55.24%. Globally, the number of cases surged from 180,018,233 (95% CI: 158,660,995–199,950,073) in 1990–324,139,599 (95% CI: 287,693,229–358,912,516) in 2021, an overall increase of 80.06%. The global age - standardized incidence rate (ASIR) grew from 3,739.9 per 100,000 population in 1990 (95% CI: 3,314.2–4,142.3) to 3,881.9 per 100,000 in 2021 (95% CI: 3,445.6–4,299.9). In contrast, China's ASIR declined from 1,851.8 per 100,000 (95% CI: 1,617.2–2,089.3) to 1,844.3 per 100,000 (95% CI: 1,605.1–2,084.1). The AAPC of incidence in China was 0.0039% (95% CI: − 0.0785 to 0.0863%), while the global AAPC for incidence was 0.1214% (95% CI: 0.1115 to 0.1313%) (S1 Table).

**Prevalence.** Regarding prevalence, the number of GERD cases in China increased from 50,632,181 (95% CI: 44,030,373–58,122,909) in 1990–81,327,260 (95% CI: 70,795,879–91,949,301) in 2021, a cumulative growth of 60.62%. Globally, the prevalence rose from 450,765,455 (95% CI: 397,478,515–511,638,410) in 1990–825,603,654 (95% CI: 732,989,500–925,555,128) in 2021, a cumulative increase of 83.16%. The global age - standardized prevalence rate (ASPR) increased from 9,516.5 per 100,000 in 1990 (95% CI: 8,427.3–10,664.7) to 9,838.6 per 100,000 in 2021 (95% CI: 8,732.5–11,056.1). Conversely, China's ASPR decreased from 4,562.6 per 100,000 (95% CI: 3,972.1–5,176.2) to 4,540.7

per 100,000 (95% CI: 3,950.8–5,156.6). The AAPC for prevalence in China was 0.0037 (95% CI: – 0.0880–0.0955%), compared with the global AAPC of 0.1098% (95% CI: 0.0963–0.1234%) (S1 Table).

**DALYs.** For DALYs, the burden of GERD in China saw the number of cases rise from 393,858 (95% CI: 199,884–694,580) in 1990–626,248 (95% CI: 311,714 − 1,115,892) in 2021, a 60.62% cumulative increase. Globally, the number of DALYs increased from 3,472,703 (95% CI: 1,752,693–6,128,627) in 1990–6,336,162 (95% CI: 3,189,794–11,241,352) in 2021, an 83.16% cumulative rise. The global age - standardized DALYs rate (ASDR) increased from 73 per 100,000 in 1990 (95% CI: 36.7–129.7) to 75.6 per 100,000 in 2021 (95% CI: 38.1–133.9), whereas China's ASDR decreased marginally from 35.3 per 100,000 (95% CI: 17.7–62.8) to 35.1 per 100,000 (95% CI: 17.6–62.3). The AAPC for DALYs in China was 0.0046% (95% CI: – 0.0861–0.0954%), while the global AAPC was 0.1129% (95% CI: 0.0991–0.1267%) (S1 Table). Geographical stratification further revealed that China's ASIR, age – standardized DALYs rate, and other relevant metrics were consistently lower than global averages (Table 1, S2 Table).

**Trends in the disease burden of GERD in China and globally from 1990 to 2021**

**Disease burden trends.** Between 1990 and 2000, the ASIR, ASDR, and ASPR of GERD in China demonstrated a downward trajectory. From 2000 to 2005, these rates remained relatively stable. Subsequently, from 2005 to 2010, there was another decrease, and from 2010 to 2021, an upward trend was evident. This pattern paralleled the global trend, suggesting that over the past decade, both the global and Chinese burden of GERD has been on the rise (Fig 1, S3 Table).

**Gender-related trends.** From 1990 to 2021, in both China and globally, the number of new GERD cases, the overall number of cases, and the number of DALYs increased over time for both males and females. However, a notable disparity was observed, with the number of cases and the corresponding rates being significantly higher in females compared to

**Table 1. All-age cases and age-standardized incidence, prevalence, and DALYs rates and corresponding AAPC of GERD in 2 and 1 in 1990 and 2021.**

| Location | Measure | 1990 | | 2021 | | 1990–2021 Percent age change, %(95%UI) | 1990-2021 Average Annual Percent Change (AAPC) |
|---|---|---|---|---|---|---|---|
| | | All-ages cases | Age-standardized rates per 100,000 people | All-ages cases | Age-standardized rates per 100,000 people | | |
| | | n (95% CI) | n (95% CI) | n (95% CI) | n (95% CI) | | |
| 2 | Incidence | 20863747 (18034679, 23688721) | 1851.8 (1617.2,2089.3) | 32387866 (27851900, 36711150) | 1844.3 (1605.1, 2084.1) | −0.4 (−0.7,-0.1) | 0.0039 (−0.0785-0.0863) |
| | Prevalence | 50632181 (44030373, 58122909) | 4562.6 (3972.1,5176.2) | 81327260 (70795879, 91949301) | 4540.7 (3950.8 ,5156.6) | −0.5 (−0.8,-0.1) | 0.0037 (−0.0880-0.0955) |
| | DALYs | 393858 (199884, 694580) | 35.3 (17.7,62.8) | 626248 (311714, 1115892) | 35.1 (17.6, 62.3) | −0.4 (−1,0.1) | 0.0046 (−0.0861-0.0954) |
| 1 | Incidence | 180018233 (158660995, 199950073) | 3739.9 (3314.2,4142.3) | 324139599 (287693229, 358912516) | 3881.9 (3445.6, 4299.9) | 3.8 (3.1,4.5) | 0.1214 (0.1115-0.1313) |
| | Prevalence | 450765455 (397478515, 511638410) | 9516.5 (8427.3,10664.7) | 825603654 (732989500, 925555128) | 9838.6 (8732.5, 11056.1) | 3.4 (2.7,4) | 0.1098 (0.0963-0.1234) |
| | DALYs | 3472703 (1752693, 6128627) | 73 (36.7,129.7) | 6336162 (3189794, 11241352) | 75.6 (38.1, 133.9) | 3.5 (2.8,4.2) | 0.1129 (0.0991-0.1267) |

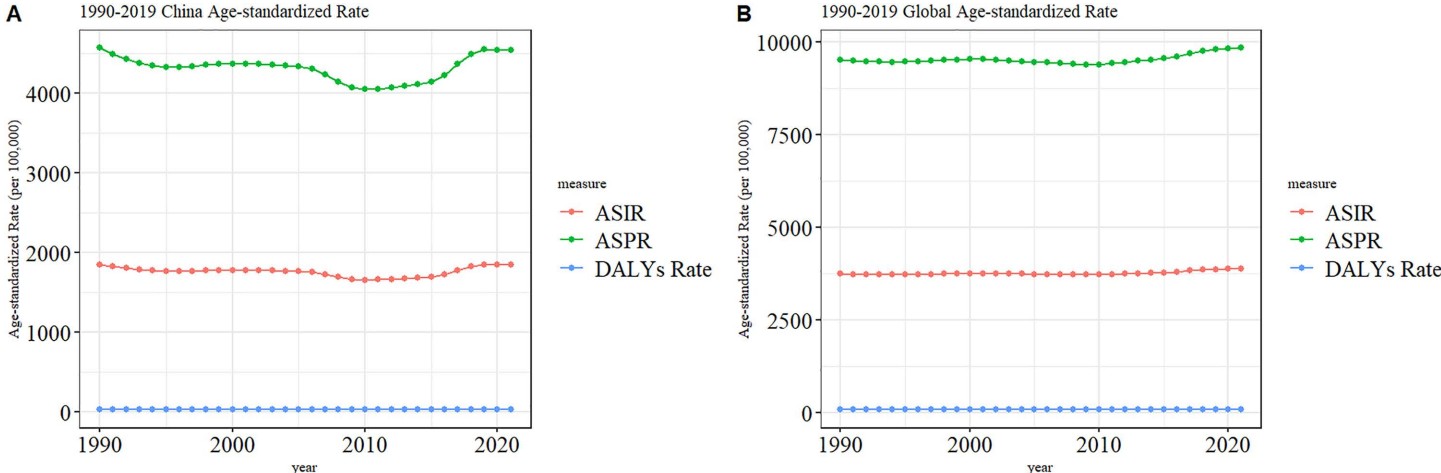

**Fig 1. Trend comparison of ASIR, ASPR, and DALYs Rate of GERD in China and Global from 1990 to 2021.**

males (Fig 2). This indicates that gender plays a role in the burden of GERD, with women being more affected. Further research into the underlying factors contributing to this gender difference in GERD burden is warranted to develop more targeted prevention and treatment strategies.

## Gender differences in the burden of GERD in different age groups in China and globally from 1990 to 2021

**Age-specific incidence, prevalence and DALYs rates in 2021.......** In 2021, in China, the peak incidence rates, prevalence rates, and DALYs rates of GERD were found in the 55–59 age group for women and the 50–54 age group for men. In contrast, globally, the peaks of these rates were observed in the 30–39 age group. Notably, in all age groups, the incidence rates of GERD were higher in women compared to men.

**Age-related trends in rates by gender.** As age increased, globally, the incidence rate, prevalence rate, and DALY rate for women reached their peaks in the 70–74 age group. For men, the prevalence rate and DALY rate peaked in the 75–79 age group, and the incidence rate peaked in the 70–74 age group. Additionally, across all age groups worldwide, the rates for women were higher than those for men in the age group under 90. In China, in all age groups, the GERD-related rates for women were higher than those for men (Fig 3, S4 Table). These findings highlight the distinct patterns of GERD burden by gender and age, which are crucial for formulating targeted public health strategies to address this condition.

## Joinpoint regression analysis of GERD burden in China and globally from 1990 to 2021

Between 1990 and 2021, the trends in ASIR, ASPR, and ASDR of GERD in China displayed multi-phase variations, with significant declines during 1990–1994 and 2001–2010, followed by periods of increase from 1994–2001, 2010–2015, 2015–2019, and 2019–2021 (all segments P < 0.05). Globally, similar alternating patterns of decrease and increase were observed over corresponding intervals, though the magnitude and timing of changes differed (Fig 4).

Notably, the overall age-standardized rates in China remained relatively stable across the entire period, reflected in near-zero average annual percentage changes (AAPCs), whereas significant increases were observed globally. This suggests that while short-term fluctuations occurred in both populations, the net disease burden of GERD in China changed minimally over three decades, contrasting with a clear upward trajectory worldwide.

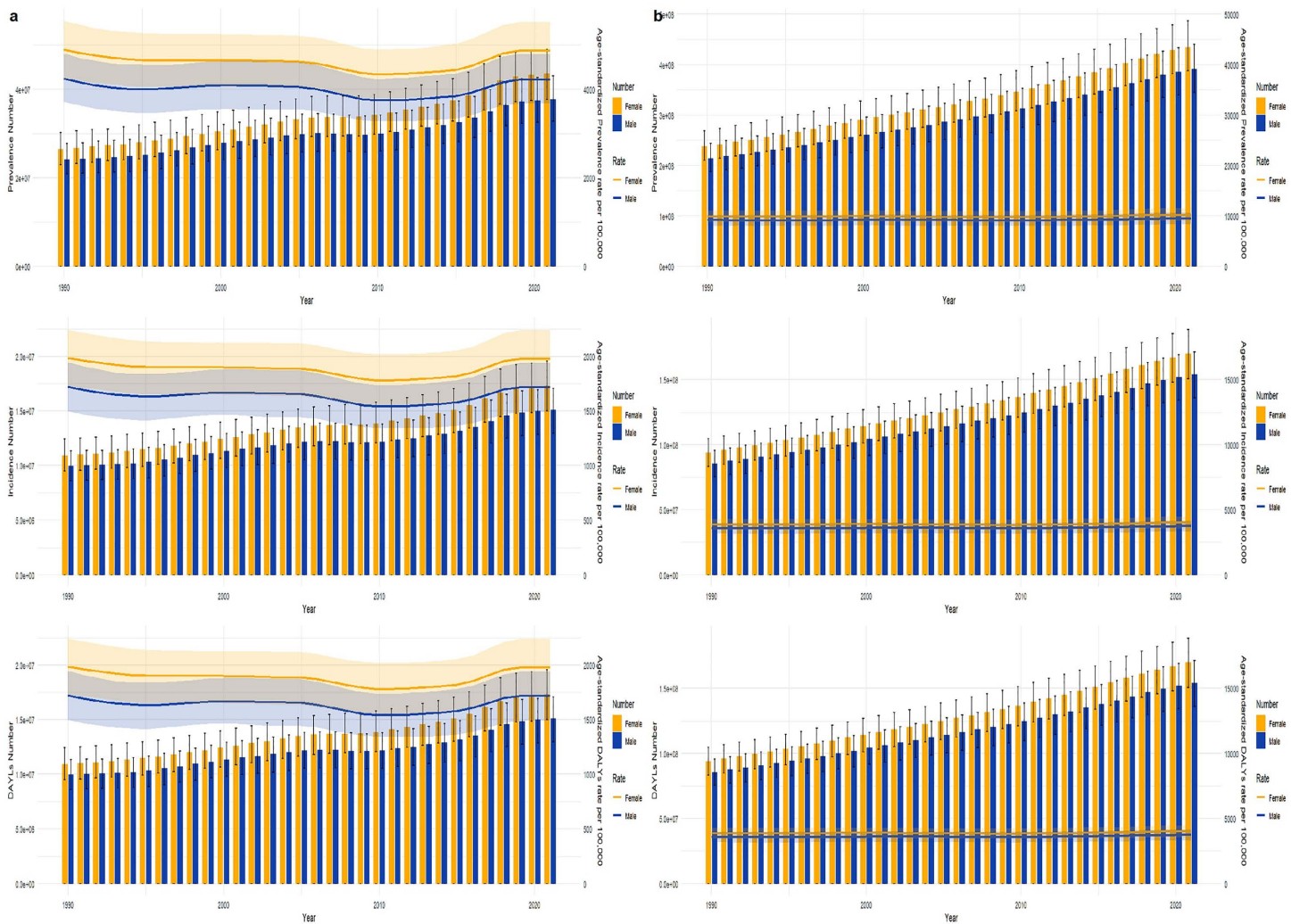

**Fig 2. Comparison of full-age cases and age-standardized rates of incidence, prevalence and DALYs among male and female in China and Global from 1990-2021.**

A detailed breakdown of APC values for each segment and statistical comparisons are provided in S5 Table and S6 Table.

## Decomposition Analysis of the Burden of GERD in China and Global from 1990 to 2021

The decomposition analysis indicates that the burden of GERD, gauged by prevalence, incidence, and DALYs, has witnessed a remarkable rise from 1990 to 2021. In 2021, globally, the annual number of GERD cases exceeded 354.4 million, with 137.6 million new cases and 2.7 million DALYs attributable to GERD. In China, there were over 29.8 million annual GERD cases in 2021, including 11.3 million new cases and 200,000 DALYs related to GERD.

Population growth emerged as the primary driver for the increase in GERD indicators both globally and in China. Over the 32-year period, population growth accounted for a 77.83% increase in the global incidence rate, a 76.42% increase in the global prevalence rate, and a 75.43% increase in global DALYs. In China, the incidence rate increased by 56.02%, the

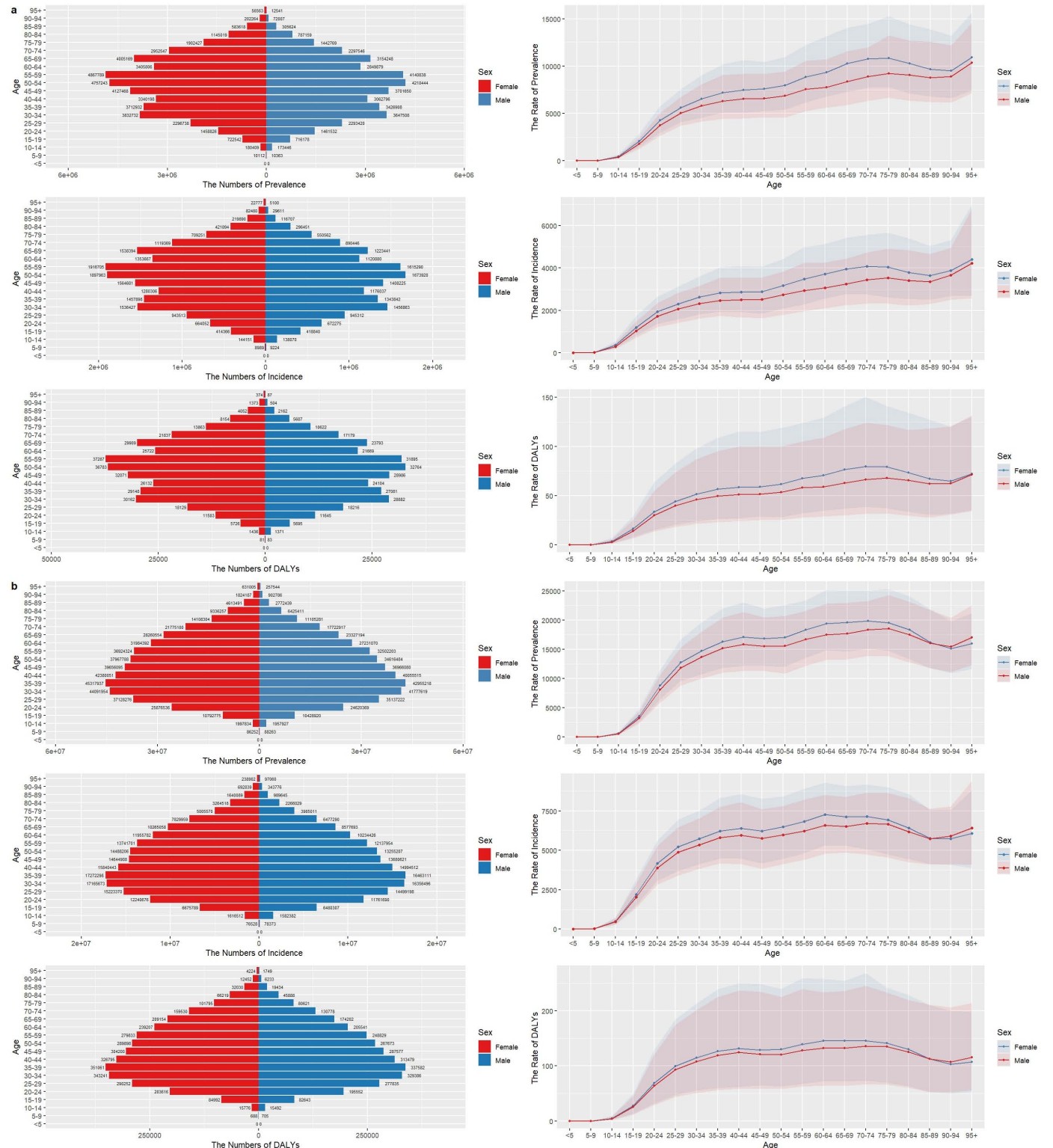

**Fig 3. Comparison of full-age cases and age-standardized rates of incidence, prevalence and DALYs among male and female in China and Global from 1990-2021.**

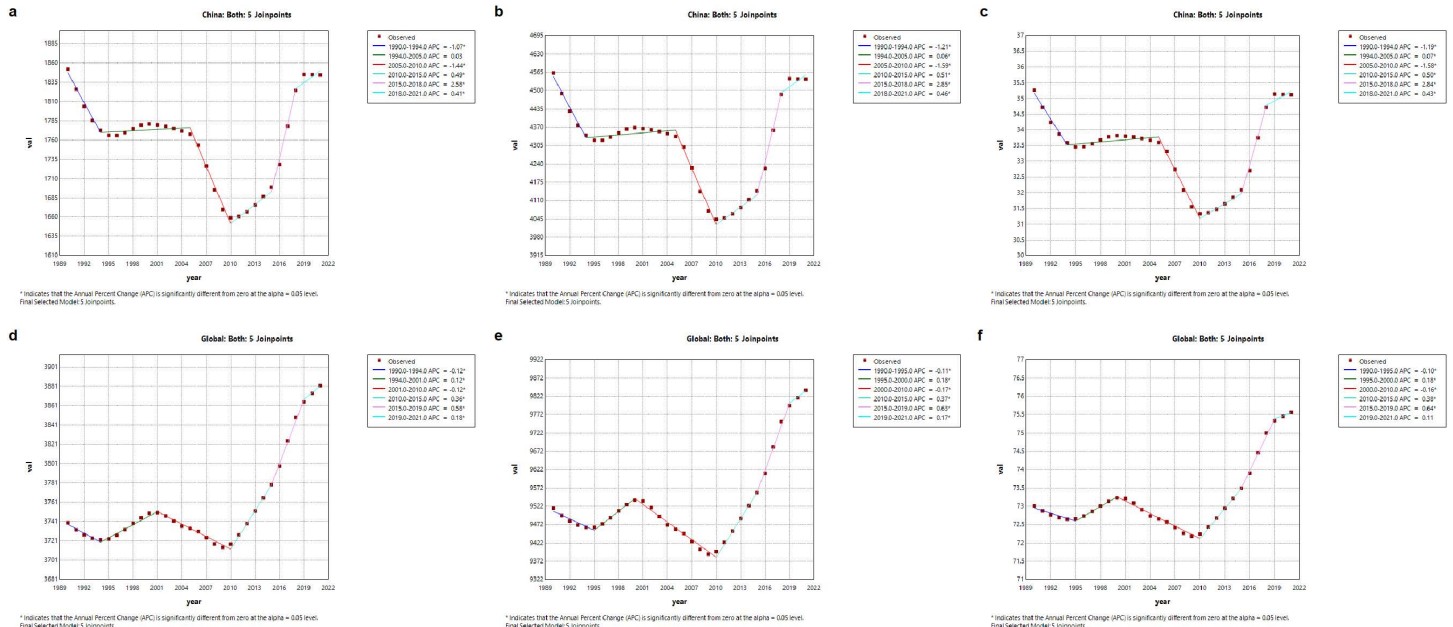

**Fig 4. Trends in ASIR, ASPR and DAYLs Rate in China and Global. (a)** The ASIR of China **(b)** The ASPR of China **(c)** The DAYLs Rate of China **(d)** The ASIR of Global **(e)** The ASPR of Global **(f)** The DAYLs Rate of Global.

prevalence rate by 50.36%, and the DALYs by 58.2%. Notably, the 50.66% increase in China's incidence rate was mainly attributed to population aging and demographic structure changes (Fig 5, S7 Table).

## Predictive analysis of the GERD burden in China by 2040

The epidemiological characteristics of GERD—including its prevalence, incidence, and DALYs—exhibit considerable variation across time and geographic regions. Conducting high-quality epidemiological studies is essential to accurately ascertain the true disease burden of GERD in different countries.

Projection results (Fig 6) indicate that by 2040, the incidence, prevalence, and DALY rates of GERD in China are expected to remain largely consistent with current levels. Specifically, the incidence rate is projected to be 2,049.32 per 100,000 people, reflecting a slight decline of 0.04%. The prevalence rate is estimated to reach 5,064.71 per 100,000, with a minor increase of 0.33%, while the DALY rate is anticipated to decrease by 2.67%, to 38.00 per 100,000.

When disaggregated by gender, male incidence is projected to rise by 0.34% to 1,913.65 per 100,000, with prevalence increasing by 0.89% to 4,726.88 per 100,000, and DALYs declining by 2.67% to 35.54 per 100,000. In contrast, females are expected to experience a slight decline in incidence (–0.05%) to 2,185.55 per 100,000, a decrease in prevalence (–0.27%) to 5,401.91 per 100,000, and a reduction in DALYs (–2.91%) to 40.55 per 100,000 (S8 Table).

Overall, although the total DALY burden of GERD in China is projected to decline from 2022 to 2040, both incidence and prevalence are predicted to rise. Notably, males are expected to bear a disproportionately higher increase in disease burden compared to females, suggesting that Chinese men may face a relatively greater challenge in managing GERD in the coming decades.

## Discussion

This study utilized data from the GBD 2021 to comprehensively characterize long-term trends in the incidence, prevalence, and DALYs of GERD in China and globally from 1990 to 2021. Overall, the global burden of GERD demonstrated

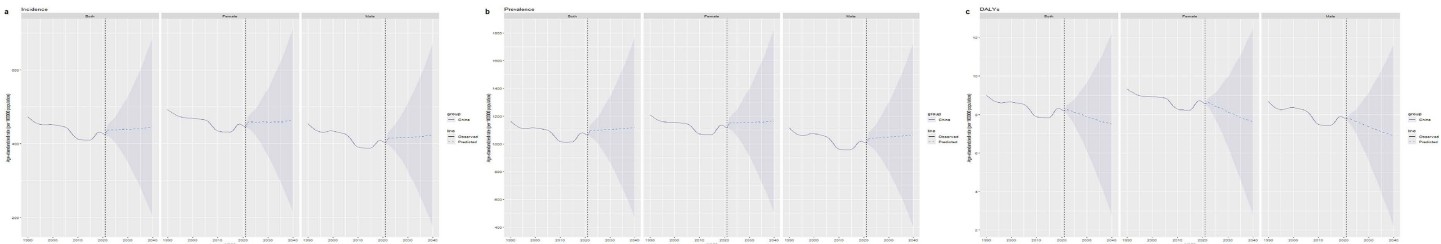

**Fig 5. Decomposition analysis of GERD indicators from 1990 to 2021 in China and Global. (a)** The Incidence of China **(b)** The Prevalence of China **(c)** The DAYLs of China **(d)** The Incidence of Global **(e)** The Prevalence of Global **(f)** The DAYLs of Global. The black dot represents the overall change value of population growth, aging, and epidemiological Change.

**Fig 6. Projection of Age-Standardized Incidence, Prevalence and DAYLs Rates of Gastroesophageal Reflux Disease in China from 2022 to 2040 Based on BAPC Modeling.**

a sustained upward trend, whereas age-standardized rates in China remained relatively stable over the study period. Despite this stability, the absolute number of GERD cases and related DALYs in China increased substantially, largely driven by population growth and aging. Pronounced heterogeneity was observed across age and sex strata, indicating that demographic factors play a central role in shaping GERD burden. These findings underscore the necessity of demographic-specific analyses when interpreting temporal trends and formulating prevention strategies.

A consistent sex-based disparity was observed, with females exhibiting higher incidence, prevalence, and DALY rates of GERD than males in both China and the global population. Hormonal factors are considered a major contributor, as estrogen has been shown to reduce lower esophageal sphincter (LES) pressure and impair esophageal motility, thereby increasing susceptibility to reflux [24]. This mechanism is further supported by evidence linking GERD risk to pregnancy, menopause, and hormone replacement therapy [25–28].In addition to biological factors, women tend to report gastrointestinal symptoms more frequently and demonstrate greater health awareness, which may contribute to higher healthcare utilization and diagnosis rates [29].Diagnostic bias and heightened clinical vigilance toward female patients may also partially explain this observed disparity [30].

A particularly notable finding of this study is the substantial difference in peak age of GERD burden between China and the global population. In China, incidence, prevalence, and DALYs peaked predominantly in individuals aged 50–59 years, whereas globally these indicators reached their highest levels in the 30–39-year age group. This discrepancy likely reflects differences in population aging trajectories, lifestyle transitions, and cumulative exposure to risk factors across regions. In China, the delayed peak may be associated with long-term dietary patterns, progressive metabolic changes, and age-related degeneration of esophageal defense mechanisms [31,32].Conversely, earlier global peaks may be driven by higher prevalence of obesity, sedentary lifestyles, and earlier exposure to dietary and psychosocial stressors in many Western countries. These findings highlight that GERD epidemiology is strongly context-dependent and should not be generalized across populations.

Age-related physiological changes further contribute to the observed burden of GERD, particularly among middle-aged and older adults. Degeneration of the anti-reflux barrier, reduced esophageal clearance, delayed gastric emptying, and increased prevalence of hiatal hernia collectively elevate reflux risk with advancing age [31,32]. Previous studies have reported higher rates of GERD-related complications, such as esophageal strictures and perforations, among individuals over 60 years of age [33].In the present study, these mechanisms likely explain the paradoxical pattern observed in China, whereby incidence and prevalence increased while DALYs declined. This trend suggests improved symptom control and reduced disability severity per case, despite a growing number of affected individuals.

Lifestyle and psychosocial factors also play an important role in shaping GERD burden. High-fat and high-sugar diets stimulate gastric acid secretion and reduce LES tone, whereas low dietary fiber intake impairs gastric emptying and glycemic control [34,35].Alcohol consumption and tobacco smoking further exacerbate reflux through delayed acid clearance and mucosal injury [36–40].Treatment outcomes are also influenced by medication adherence, as inappropriate dose adjustment and poor compliance may compromise therapeutic efficacy and accelerate disease progression [41–44]. however, in China, strengthened primary healthcare services and improved accessibility of proton pump inhibitors may partially mitigate these effects. In addition, occupational stress and chronic psychological distress have been implicated in GERD pathogenesis through alterations in gastrointestinal motility and inflammatory pathways, with women potentially being more susceptible due to higher prevalence of anxiety and depression [25,45–48].

Importantly, the absence of clearly defined age- and sex-specific epidemiological trends presents a significant obstacle to the effective management of RGERD. RGERD is increasingly recognized as a heterogeneous condition influenced by hormonal status, visceral sensitivity, psychosocial stress, and age-related physiological changes [49,1]. However, current treatment strategies remain largely uniform across demographic groups [50]. Without adequate demographic stratification, clinicians may fail to identify high-risk subpopulations who are more likely to respond poorly to standard-dose proton pump inhibitors [50]. This limitation hinders early therapeutic optimization and constrains the development of individualized management strategies, ultimately compromising clinical outcomes.

Advances in diagnostic technologies, including high-resolution esophageal manometry and 24-hour multichannel intraluminal impedance–pH monitoring, have substantially improved the accuracy of GERD diagnosis and phenotyping [51–55]. These tools enable more precise assessment of reflux patterns and esophageal function, facilitating personalized treatment approaches. In parallel, improved access to healthcare services and widespread availability of proton pump

inhibitors in China may have contributed to reduced disease severity, as reflected by declining DALYs. Nevertheless, the continued rise in incidence and prevalence highlights the need for sustained prevention efforts and early intervention.

Taken together, these findings emphasize that GERD burden is shaped by complex interactions among demographic structure, biological mechanisms, lifestyle factors, and healthcare systems. The pronounced heterogeneity across age and sex groups underscores the necessity of incorporating demographic-specific evidence into clinical guidelines and public health policies. Targeted screening of high-risk populations, standardized diagnostic criteria, and tailored management strategies are urgently needed. Furthermore, international collaboration is essential to improve cross-national comparability and advance precision public health approaches to GERD prevention and control.

## Study limitations

First, data-related biases may originate from the GBD database and regional registry systems. For instance, in regions with limited access to endoscopic facilities, the prevalence of certain conditions —including GERD— may be underestimated due to insufficient diagnostic capabilities. Conversely, in wealthier areas with widespread use of PPIs, GERD-related symptoms may be masked, thereby distorting actual disease burden estimates. Moreover, inconsistencies in diagnostic criteria for GERD across different countries further challenge the validity of cross-national comparisons. These limitations are particularly relevant when comparing trends between China and other regions, as disparities in health infrastructure and clinical practices may lead to erroneous interpretations of differential disease burdens.

Second, the lack of detailed, micro-level data hampers the ability to identify differences in disease patterns between developed and remote areas. A more thorough analysis of factors such as educational attainment, socioeconomic status, dietary habits, and local environmental exposures—and their interactions with GERD—would necessitate higher-resolution data. Future studies should prioritize the collection of such individualized and region-specific data to better understand the determinants and heterogeneities of GERD epidemiology.

Third, this study did not incorporate extensive cross-regional or cross-country comparisons that might reveal important variations in GERD burden, such as between Eastern and Western China or across countries with similar economic development levels. Conducting such comparative analyses could offer valuable insights into the effectiveness of different public health interventions and policies.

Therefore, future research should aim to enhance the coverage and quality of GERD registries, especially in rural and low-income settings. Additionally, implementing well-designed longitudinal cohort studies could help elucidate causal relationships and mechanistic pathways. Furthermore, international collaborations aiming to standardize diagnostic criteria and data collection protocols across regions are essential to improve the comparability and reliability of future burden estimates.

## Conclusion

As evidenced in this study, GERD represents a growing and distinctive public health challenge in China, exhibiting trends that diverge from global patterns. Although the DALYs related to GERD are projected to decline, both incidence and prevalence are expected to rise—primarily driven by demographic aging and population growth. These trends occur within a complex framework of interacting risk factors, including mental health status, disparities in healthcare access, socioeconomic conditions, and overall quality of life.

A key and novel finding of this study is the rapidly increasing disease burden among males, which—although current rates remain higher in females—is projected to surpass the burden in women in the near future. This shift underscores the urgent need for gender-specific and age-tailored public health interventions, particularly aimed at male and older adult populations within China.

Furthermore, the persistent disparities in diagnostic practices and criteria across regions highlight the necessity of cross-national comparative studies. Such efforts are critical to validate observed differences, align diagnostic protocols,

and facilitate the development of targeted strategies that account for regional variability in healthcare capacity and social determinants of health.

In light of these findings, we strongly advocate for the implementation of precision public health measures that respond to these demographic and gendered trends, alongside promoting international collaboration to advance equitable GERD management and policy formulation.

## Supporting information

**S1 Table. The global prevalencecountries and territories in 202cidence and DAYLs burden of GERD in 204 countries and territories in 2021.**
(XLSX)

**S2 Table. All-age cases and age-standardized incidence, prevalence, and DALYs rates and corresponding AAPC of GERD in China and globally in 1990 and 2021.**
(XLSX)

**S3 Table. Trend comparison of ASIR, ASPR, and DALYs Rate of GERD in China and Global from 1990 to 2021.**
(XLSX)

**S4 Table. Comparison of full-age cases and age-standardized rates of incidence, prevalence and DALYs among male and female in China and Global from 1990–2021.**
(XLSX)

**S5 Table. The age-period-cohort of incidence, Prevalence and DAYLs in china and global from 1990 to 2021.**
(XLSX)

**S6 Table. Trends in ASIR, ASPR and DAYLs Rate in China and Global from 1990 to 2021.**
(XLSX)

**S7 Table. Decomposition analysis of GERD indicators from 1990 to 2021 in China and Global.**
(XLSX)

**S8 Table. Trends in ASIR, ASPR and DAYLs Rate in China from 1990 to 2040.**
(XLSX)

## Acknowledgments

We truly appreciate the efforts of the Global Burden of Disease Study 2021 collaborators in delivering the most complete study of various diseases on a worldwide scale. We also express our sincere appreciation to the Institute for Health Metrics and Evaluation (IHME) for making the GBD data available for this research.

## Author contributions

**Conceptualization:** zexu tang, E Xie.

**Data curation:** Jinwen Fang.

**Investigation:** Yongjian Huang.

**Methodology:** Zelin Chen.

**Software:** Yongjian Huang.

**Supervision:** Chunzhu Li.

**Validation:** Xiangyang Yang.

**Writing – original draft:** zexu tang, Huiying Zhang.

**Writing – review & editing:** zexu tang, E Xie.

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
