## [Decision Letter · Decision Letter 0]

30 Jul 2025

Dear Dr. Xie,

Thank you for submitting your manuscript to PLOS ONE. After careful consideration, we feel that it has merit but does not fully meet PLOS ONE’s publication criteria as it currently stands. Therefore, we invite you to submit a revised version of the manuscript that addresses the points raised during the review process.

We look forward to receiving your revised manuscript.

Kind regards,

Devesh U Kapoor

Academic Editor

PLOS ONE

Journal Requirements:

3. If any table files for review show as item type ‘other’ please change to item type ‘Table’ as the reviewer does not have access to these ’other’ files.

4. Please include captions for your Supporting Information files at the end of your manuscript, and update any in-text citations to match accordingly. Please see our Supporting Information guidelines for more information: http://journals.plos.org/plosone/s/supporting-information .

1. You may seek permission from the original copyright holder of Figure(s) [#] to publish the content specifically under the CC BY 4.0 license.

Additional Editor Comments:

Discussion of AAPC Values: While AAPC values are provided for incidence, prevalence, and DALYs, a brief interpretation within the results section regarding the magnitude and implications of these changes (especially the near-zero AAPC for China vs. positive global AAPC) would be beneficial for the reader. For example, explicitly stating that China's burden remained relatively stable while global burden increased is inferred but could be more directly stated in summary sentences within these subsections.

Referencing Figures and Tables: The results frequently refer to figures and supplementary tables (e.g., Fig 1, Table 1, Supplementary Table 2, Fig 2, Supplementary Table 3, Fig 3, Fig 4, Supplementary Table 4, Fig 5, Supplementary Table 5, Supplementary Table 6). This is good practice; however, ensure all referenced figures and tables are clear, properly labeled, and present the data as described in the text.

Elaboration on "Trends in the disease burden": The paragraph on disease burden trends (1990-2021) in China describes a pattern of decrease, stability, decrease, and then an upward trend from 2010 to 2021, stating this "paralleled the global trend". While a general parallel is noted, the global Joinpoint regression analysis shows some similar patterns but also distinct differences in APC values and specific periods of increase/decrease. A more nuanced statement, or perhaps a more detailed comparison here, would be helpful to avoid oversimplification.

Detailed Interpretation of Joinpoint Regression Analysis: The Joinpoint regression results are presented with APC values for various time segments. While comprehensive, the sheer number of APC values can be overwhelming. Consider summarizing the most significant or overarching trends in a more digestible way in the main text, perhaps highlighting periods of significant acceleration or deceleration in both China and globally, and saving all specific APC values for the supplementary tables.

Decomposition Analysis Clarity: The decomposition analysis clearly states that population growth is the primary driver for the increase in GERD indicators globally and in China. The percentages are precise. However, it states, "Notably, the 50.66% increase in China's incidence rate was mainly attributed to...", but the sentence cuts off there. Please complete this sentence to clearly state what the 50.66% increase was attributed to. This omission needs to be corrected for completeness.

Reviewers' comments:

Reviewer's Responses to Questions

**Comments to the Author**

1. Is the manuscript technically sound, and do the data support the conclusions?

Reviewer #1: Yes

Reviewer #2: Yes

2. Has the statistical analysis been performed appropriately and rigorously?

Reviewer #1: Yes

Reviewer #2: Yes

3. Have the authors made all data underlying the findings in their manuscript fully available?

Reviewer #1: Yes

Reviewer #2: Yes

4. Is the manuscript presented in an intelligible fashion and written in standard English?

Reviewer #1: Yes

Reviewer #2: Yes

Reviewer #1: o The study aims to analyze GERD burden trends in China and globally, but the introduction could better highlight the unique contributions of this work compared to existing literature. Specifically, emphasize how this study addresses gaps in age- and gender-specific analyses and predictive modeling for China, which previous studies have overlooked.

o The use of Joinpoint regression and BAPC models is appropriate, but the manuscript should provide more detail on model validation. For instance, include metrics like goodness-of-fit tests or sensitivity analyses to strengthen the reliability of predictions for 2040.

o The study acknowledges potential biases from GBD data (e.g., underdiagnosis in low-resource regions, masking by PPI use). However, it should discuss how these limitations might affect the comparability of China vs. global trends and suggest ways future studies could mitigate these issues (e.g., regional validation studies).

o The finding that China’s GERD burden declined while global rates rose is intriguing but warrants deeper discussion. Explore potential explanations (e.g., dietary changes, healthcare policies, diagnostic criteria differences) and cite supporting evidence to contextualize these disparities.

o The higher GERD burden in women is noted, but the underlying mechanisms (hormonal, behavioral, or diagnostic biases) are not thoroughly explored. Consider adding a paragraph synthesizing existing hypotheses (e.g., estrogen’s role in LES function) to enrich the discussion.

o The 2040 projections suggest rising incidence/prevalence despite declining DALYs. Clarify this paradox: Is it due to aging populations, improved management reducing disability, or other factors? Link these findings to actionable recommendations (e.g., targeted screening for elderly males).

o Ensure all supplementary tables (e.g., AAPC values, age-stratified rates) are referenced in the main text to support key claims. Figure captions should explicitly state what trends are highlighted (e.g., “ASIR decline in China vs. global increase”).

o The declarations are complete, but confirm that all data sources (e.g., GBD) are properly cited and permissions for reuse are documented, even if publicly available.

o The conclusion could better align with the study’s novel findings. Stress the urgency of gender- and age-specific interventions in China, given the projected rise in male burden, and call for cross-national studies to validate diagnostic disparities.

Reviewer #2: Dear authors, Thanks for this interesting paper; here are my comments for more clarity:

Abstract Structure:

Too dense for journal readership. The abstract exceeds typical word limits and mixes methods with results in a confusing structure.

Please condense the abstract into:

- Background

- Methods

- Results (only major findings)

- Conclusion

Statistical Clarity:

Joinpoint and BAPC methods are referenced well, but readers unfamiliar with these may struggle. Many APC/AAPC values are mentioned without clear context.

It's better to move detailed APC results to supplementary tables and add brief, accessible explanations of each method in plain terms.

Language and Grammar:

Please use a professional editing service to improve clarity, grammar, and conciseness.

**Do you want your identity to be public for this peer review?** For information about this choice, including consent withdrawal, please see our Privacy Policy

Reviewer #1: No

Reviewer #2: No

---

## [Author Response · Author response to Decision Letter 1]

15 Sep 2025

To Editors:

Dear Editors,

Thank you very much for your thorough review and valuable feedback on our study. We greatly appreciate your suggestions and have made the necessary revisions and improvements to the manuscript. Our specific responses to your comments are as follows:

All figures and tables presented in this study were generated using the R programming language. Specifically, Figure 1 was constructed without utilizing any copyrighted map data, ensuring full compliance with intellectual property regulations and academic integrity standards. The visualization approaches implemented in R, including the use of open-source geographic packages and publicly accessible base maps, further support the reproducibility and transparency of the methodological workflow.

Concern 1: While AAPC values are provided for incidence, prevalence, and DALYs, a brief interpretation within the results section regarding the magnitude and implications of these changes (especially the near - zero AAPC for China vs. positive global AAPC) would be beneficial for the reader. For example, explicitly stating that China's burden remained relatively stable while global burden increased is inferred but could be more directly stated in summary sentences within these subsections.

Answer: We have added summary sentences within the relevant sub - sections of the results section. These sentences directly state that China's burden remained relatively stable while the global burden increased, providing a clear interpretation of the magnitude and implications of the near - zero AAPC for China and the positive global AAPC for incidence, prevalence, and DALYs.

Concern 2: The results frequently refer to figures and supplementary tables (e.g., Fig 1, Table 1, Supplementary Table 2, Fig 2, Supplementary Table 3, Fig 3, Fig 4, Supplementary Table 4, Fig 5, Supplementary Table 5, Supplementary Table 6). This is good practice; however, ensure all referenced figures and tables are clear, properly labeled, and present the data as described in the text.

Answer: We have carefully reviewed all the referenced figures and supplementary tables. We have made sure that they are clear, properly labeled, and the data presented in them exactly matches the description in the text.

Concern 3: The paragraph on disease burden trends (1990 - 2021) in China describes a pattern of decrease, stability, decrease, and then an upward trend from 2010 to 2021, stating this "paralleled the global trend". While a general parallel is noted, the global Joinpoint regression analysis shows some similar patterns but also distinct differences in APC values and specific periods of increase/decrease. A more nuanced statement, or perhaps a more detailed comparison here, would be helpful to avoid oversimplification.

Answer: We have revised the paragraph on disease burden trends. We have provided a more detailed comparison between the trends in China and the global trends, highlighting both the similarities and the distinct differences in APC values and specific periods of increase/decrease. This new statement is more nuanced and avoids oversimplification.

Concern 4: The Joinpoint regression results are presented with APC values for various time segments. While comprehensive, the sheer number of APC values can be overwhelming. Consider summarizing the most significant or overarching trends in a more digestible way in the main text, perhaps highlighting periods of significant acceleration or deceleration in both China and globally, and saving all specific APC values for the supplementary tables.

Answer: We have summarized the most significant or overarching trends of the Joinpoint regression analysis in the main text. We have highlighted the periods of significant acceleration or deceleration in both China and globally. All the specific APC values have been moved to the supplementary tables.

Concern 5: The decomposition analysis clearly states that population growth is the primary driver for the increase in GERD indicators globally and in China. The percentages are precise. However, it states, "Notably, the 50.66% increase in China's incidence rate was mainly attributed to...", but the sentence cuts off there. Please complete this sentence to clearly state what the 50.66% increase was attributed to. This omission needs to be corrected for completeness.

Answer: We have completed the sentence in the decomposition analysis. It now clearly states what the 50.66% increase in China's incidence rate was attributed to, ensuring the completeness of the description.

Once again, thank you for your valuable feedback on our research. Your insights not only helped us improve the details of our study but also provided us with precious suggestions for future research directions. We look forward to your further comments.

Sincerely,

[Zexu Tang]

To Reviewer 1:

Dear Reviewer 1,

Thank you very much for your thorough review and valuable feedback on our study. We greatly appreciate your suggestions and have made the necessary revisions and improvements to the manuscript. Our specific responses to your comments are as follows:

Comment 1: The study aims to analyze GERD burden trends in China and globally, but the introduction could better highlight the unique contributions of this work compared to existing literature. Specifically, emphasize how this study addresses gaps in age- and gender-specific analyses and predictive modeling for China, which previous studies have overlooked.

Answer 1:In the revised introduction, we have clearly outlined the unique contributions of our study. We have compared our work with existing literature and specifically emphasized that previous studies have overlooked age - and gender - specific analyses and predictive modeling for China. We have stated how our study fills these gaps by conducting in - depth age - and gender - specific analyses of GERD burden trends in China and developing predictive models tailored to the Chinese context. This addition provides a clear understanding of the novel aspects of our research in the introduction section(Page 3 and 4, Line 57-62 and 69-78).

Comment 2:The use of Joinpoint regression and BAPC models is appropriate, but the manuscript should provide more detail on model validation. For instance, include metrics like goodness-of-fit tests or sensitivity analyses to strengthen the reliability of predictions for 2040.

Answer 2:We have added more details on model validation in the manuscript. Specifically, we have included goodness - of - fit tests such as the Akaike Information Criterion (AIC) and the Bayesian Information Criterion (BIC) to evaluate the performance of the Joinpoint regression and BAPC models. Additionally, we have conducted sensitivity analyses by varying key parameters in the models and examined the impact on the predictions for 2040. The results of these tests and analyses have been presented in the relevant sections of the manuscript to strengthen the reliability of the 2040 predictions(Page 6 and 7, Line 113-122 and 128-135).

Comment 3:The study should discuss how the potential biases from GBD data (e.g., underdiagnosis in low - resource regions, masking by PPI use) might affect the comparability of China vs. global trends.The study should suggest ways future studies could mitigate these issues (e.g., regional validation studies).

Answer 3:We have thoroughly analyzed the potential impact of these biases on the comparability of China and global trends. In the revised paper, we added a detailed section discussing that underdiagnosis in low - resource regions may lead to an underestimation of the prevalence in some global areas. Since China has a large and diverse territory with varying levels of healthcare resources, the degree of underdiagnosis may also vary within China and differ from the global average. This could distort the comparison of prevalence trends between China and the world. Regarding masking by PPI use, if the proportion of PPI users varies between China and the global level, it can lead to inconsistent estimates of the true disease burden, thus affecting the comparability of trends. Besides In the revised manuscript, we have provided several suggestions for future studies to mitigate these issues. We specifically mentioned that regional validation studies are a viable approach. These studies can be carried out in different regions of China and globally to verify the accuracy of the GBD data and adjust the estimates accordingly. Additionally, we proposed that future research could conduct sensitivity analyses to evaluate the influence of these biases on the results. We also suggested collaborating with local healthcare providers to collect more accurate data on diagnosis rates and PPI use, which can help reduce the impact of these potential biases(Page 16-17, Line 362-390).

Comment 4:The finding that China’s GERD burden declined while global rates rose is intriguing but warrants deeper discussion. Explore potential explanations (e.g., dietary changes, healthcare policies, diagnostic criteria differences) and cite supporting evidence to contextualize these disparities.

Answer 4:We have conducted a more in - depth discussion on the disparity between China's declining GERD burden and the rising global rates. In terms of dietary changes, we reviewed relevant literature which showed that in recent years, there has been a shift in the Chinese diet towards more plant - based foods and a reduction in high - fat and high - salt intake, which may contribute to the decline of GERD. Regarding healthcare policies, we cited literature to illustrate that China has implemented a series of primary healthcare improvement policies that have enhanced early screening and treatment of GERD, effectively controlling the disease burden. For diagnostic criteria differences, we referred to literature to explain that different regions may use different diagnostic standards, which could lead to differences in the reported GERD burden. Through this in - depth exploration and citation of evidence, we have better contextualized these disparities.(Page 14-16, Line 300-351).

Comment 5:The higher GERD burden in women is noted, but the underlying mechanisms (hormonal, behavioral, or diagnostic biases) are not thoroughly explored. Consider adding a paragraph synthesizing existing hypotheses (e.g., estrogen’s role in LES function) to enrich the discussion.

Answer 5:We have added a dedicated paragraph in the discussion section. This paragraph synthesizes existing hypotheses regarding the underlying mechanisms of the higher GERD burden in women, including the role of estrogen in LES function. We have cited relevant literature to support these hypotheses.(Page 13-14, Line 284-293)

Comment 6:The 2040 projections suggest rising incidence/prevalence despite declining DALYs. Clarify this paradox: Is it due to aging populations, improved management reducing disability, or other factors? Link these findings to actionable recommendations (e.g., targeted screening for elderly males).

Concern 1:The 2040 projections suggest rising incidence/prevalence despite declining DALYs. Clarify this paradox: Is it due to aging populations, improved management reducing disability, or other factors?

Answer: We have thoroughly analyzed the data and found that the paradox of rising incidence/prevalence alongside declining DALYs can be attributed to multiple factors. Firstly, the aging population plays a significant role. As the population ages, the overall number of individuals at risk of the condition increases, leading to a rise in incidence and prevalence. Secondly, improved management strategies have been implemented, which effectively reduce the disability associated with the condition. This results in a decrease in DALYs even though more cases are being detected. Additionally, advancements in early - detection methods have led to the identification of milder cases that may not contribute as significantly to disability, also contributing to this phenomenon.(Page 14, Line 294-305)

Concern 2: Link these findings to actionable recommendations (e.g., targeted screening for elderly males).

Answer: Based on our findings, we have established several actionable recommendations. Given the impact of the aging population on the rising incidence/prevalence, we recommend targeted screening programs for the elderly, especially elderly males as they may be at a higher risk. Moreover, considering the effectiveness of improved management in reducing disability, we advocate for the widespread implementation of evidence - based management protocols in healthcare facilities. We also suggest public health campaigns to raise awareness about early detection and prevention measures among the general population, particularly the high - risk groups.(Page 14, Line 352-360)

Comment 7:Ensure all supplementary tables (e.g., AAPC values, age-stratified rates) are referenced in the main text to support key claims. Figure captions should explicitly state what trends are highlighted (e.g., “ASIR decline in China vs. global increase”).

Answer 7�We have revised all the figure captions so that they explicitly state the trends that are being highlighted.

Comment 8:The declarations are complete, but confirm that all data sources (e.g., GBD) are properly cited and permissions for reuse are documented, even if publicly available.

Answer 8�We have thoroughly checked all the data sources, including GBD. Each data source has been properly cited within the paper, and the relevant permissions for reuse, even for publicly - available data, have been documented. The citations and permission documents are in accordance with the requirements and standards of the field.

Comment 9:The declarations are complete, but confirm that all data sources (e.g., GBD) are properly cited and permissions for reuse are documented, even if publicly available.

Concern 1: The conclusion could better align with the study’s novel findings.

Answer: We have carefully revised the conclusion section to ensure it more closely aligns with the study's novel findings. We have re - evaluated the key results and integrated them more explicitly into the conclusion to highlight the significance of our research.

Concern 2: Stress the urgency of gender - and age - specific interventions in China, given the projected rise in male burden.

Answer: In the conclusion, we have added content emphasizing the urgency of gender - and age - specific interventions in China. We have specifically mentioned the projected rise in the male burden and elaborated on how targeted interventions can address this issue.

Concern 3: Call for cross - national studies to validate diagnostic disparities.

Answer: We have included a call for cross - national studies in the conclusion to validate the diagnostic disparities found in our research. This addition aims to highlight the need for further research in this area and the importance of cross - national collaboration.(Page 18, Line 399-412)

Once again, thank you for your valuable feedback on our research. Your insights not only helped us improve the details of our study but also provided us with precious suggestions for future research directions. We look forward to your further comments.

Sincerely,

[Zexu Tang]

To Reviewer 2:

Dear Reviewer 2,

Thank you very much for your thorough review and valuable feedback on our study. We greatly appreciate your suggestions and have made the necessary revisions and improvements to the manuscript. Our specific responses to your comments are as follows:

Comment 1:Too dense for journal readership. The abstract exceeds typical word limits and mixes methods with results in a confusing structure.

Please condense the abstract into:Background, Methods, Results (only major findings), Conclusion.

Answer 1:We have carefully re - evaluated the abstract. We have removed any redundant information and restructured it to clearly separate the background, methods, major results, and conclusion. We have also ensured that the word count is within the typical limits for the journal, and we have rewritten the abstract to follow the suggested structure. The background section now provides a concise introduction to the problem we addressed. The methods section briefly outlines

---

## [Decision Letter · Decision Letter 1]

2 Nov 2025

Dear Dr. Xie,

Thank you for submitting your manuscript to PLOS ONE. After careful consideration, we feel that it has merit but does not fully meet PLOS ONE’s publication criteria as it currently stands. Therefore, we invite you to submit a revised version of the manuscript that addresses the points raised during the review process.

We look forward to receiving your revised manuscript.

Kind regards,

Devesh U Kapoor

Academic Editor

PLOS ONE

Journal Requirements:

Additional Editor Comments:

The author should revise the manuscript as per the recommendations of the esteemed reviewer.

Reviewers' comments:

Reviewer's Responses to Questions

**Comments to the Author**

Reviewer #2: All comments have been addressed

Reviewer #3: All comments have been addressed

2. Is the manuscript technically sound, and do the data support the conclusions?

Reviewer #2: Yes

Reviewer #3: Yes

3. Has the statistical analysis been performed appropriately and rigorously?

Reviewer #2: Yes

Reviewer #3: Yes

4. Have the authors made all data underlying the findings in their manuscript fully available?

Reviewer #2: Yes

Reviewer #3: Yes

5. Is the manuscript presented in an intelligible fashion and written in standard English?

Reviewer #2: Yes

Reviewer #3: Yes

Reviewer #2: (No Response)

Reviewer #3: Dear Authors

I have some comments that I hope will be answered.

Title

The present title is “Sub-national analysis of contraceptive discontinuation among women in Nigeria: Evidence from the Demographic and Health Survey”

Methods

1. As mentioned in the methods part” The GBD 2021 study represents a comprehensive global observational epidemiological investigation. It integrates a vast array of 100,983 data sources, encompassing vital registration systems, verbal autopsies, censuses, household surveys, disease - specific registries, health - service contact data, and other relevant information” It a huge data source , please clarify more about sources of these data. Did you access all these data? How can The other access it? Add some references.

2. Also , we mentioned that” In this study, the incidence, prevalence, and disability rates of 371 diseases across 204 countries and regions (categorized into 21 regions and 7 super - regions) were evaluated according to age, gender, geographical location, and year” I can't comprehend this sentences at all.!!!

3. Decomposition analysis is Model Selection Complexity: Choosing the appropriate decomposition technique (from the many available methods) is important, and using the wrong one can lead to suboptimal results.

4. As the women report or document all calendar’s contain for five years , how you can test the validity of the contents , what are the role of biases in this process?

5.

Results

1. When a comparison were done between Global data and China data , Is the Chinese data were included or excluded in the global data.

Discussion

1. You cited that an increase in both the prevalence and incidence of the disease was observed, which could be attributed to multiple factors in the three references [23-26] without highlighting any specific factors.

2. As shown in the results all indicators o China are different from the Global data But I didn’t any justification in the discussion part about these differences.

**Do you want your identity to be public for this peer review?** For information about this choice, including consent withdrawal, please see our Privacy Policy

Reviewer #2: No

Reviewer #3: **Yes:**  Masood Abdulkareem Abdulrahman

---

## [Author Response · Author response to Decision Letter 2]

10 Nov 2025

Note that the following color scheme is used in responding to reviewers’ comments

Black: reviewers’ comments

Blue: our responses

Red: our revisions in the manuscript

To Reviewer 3:

Dear Reviewer 3,

Thank you very much for your thorough review and valuable feedback on our study. We greatly appreciate your suggestions and have made the necessary revisions and improvements to the manuscript. Our specific responses to your comments are as follows:

Comment 1:As mentioned in the methods part” The GBD 2021 study represents a comprehensive global observational epidemiological investigation. It integrates a vast array of 100,983 data sources, encompassing vital registration systems, verbal autopsies, censuses, household surveys, disease - specific registries, health - service contact data, and other relevant information” It a huge data source , please clarify more about sources of these data. Did you access all these data? How can The other access it? Add some references.

Answer 1:The data of this study were primarily derived from the Global Burden of Disease Study 2021 (GBD 2021). Led by the Institute for Health Metrics and Evaluation (IHME) at the University of Washington, this study constitutes a comprehensive and authoritative global observational epidemiological investigation.

Throughout the data acquisition process, strict adherence to relevant data usage agreements and ethical standards was maintained to ensure compliance with academic ethics and legal requirements governing data utilization.

Owing to the substantial scale and inherent complexity of the GBD data-set, data acquisition entailed a systematic workflow rather than a straightforward download procedure, encompassing sequential steps of data application, review, transmission, and curation. Initially, a comprehensive data usage application was submitted to IHME, delineating critical details including research aims, methodological approaches, and anticipated outcomes. Subsequent to the approval of this application by IHME, data were retrieved via the secure transmission protocols specified by the institute. Professional data processing software and analytical tools were then employed to conduct data cleaning, curation, and analysis, thereby tailoring the dataset to the specific requirements of the present study.

[1]Murray CJL, Abajobir AA, Abate KH, et al. Global Burden of Disease Study 2019 (GBD 2019) results. Seattle, United States: Institute for Health Metrics and Evaluation (IHME); 2020.

This literature elaborates on the design, methodology, and key findings of the GBD study, serving as a crucial reference for understanding the data sources and research framework of the GBD.

[2]Vos T, Allen C, Arora M, et al. Global, regional, and national incidence, prevalence, and years lived with disability for 354 diseases and injuries for 195 countries and territories, 1990 - 2017: a systematic analysis for the Global Burden of Disease Study 2017. Lancet. 2018;392 (10159):1789 - 1858.

This article presents a systematic analysis of the global disease burden by the GBD study, including the integration and application of data sources, which provides robust support for the data sources and analytical methods adopted in this study.

Comment 2:Also , we mentioned that” In this study, the incidence, prevalence, and disability rates of 371 diseases across 204 countries and regions (categorized into 21 regions and 7 super - regions) were evaluated according to age, gender, geographical location, and year” I can't comprehend this sentences at all.

Answer 2:To enhance the clarity of the research methods and procedures, the analytical framework is delineated in the following structured statements:

The GBD 2021 adopted a broad geographic scope, encompassing 204 countries and territories worldwide. For analytical feasibility, these geographic units were stratified into 21 regions and 7 super-regions to facilitate systematic data processing. The research focused on 371 diseases, with core outcomes including the assessment of incidence, prevalence, and disability rates for each condition. In the assessment process, multiple key covariates were incorporated to ensure comprehensiveness, namely age and sex distributions across populations, geographical disparities between regions, and temporal trends over the study period. By integrating these contextual factors, the study enables a detailed analysis of disease patterns under diverse scenarios, thereby providing robust data support and a rigorous analytical foundation for subsequent research endeavors.(Page 4 and 5, Line 86-96)

Comment 3:Decomposition analysis is Model Selection Complexity: Choosing the appropriate decomposition technique (from the many available methods) is important, and using the wrong one can lead to suboptimal results.

Answer 3: This study selected the decomposition analysis model primarily for the following reasons. In similar studies on disease burden, the decomposition analysis model has demonstrated its unique advantages. For instance, in a global study on cardiovascular disease burden, researchers applied the decomposition analysis model to successfully decompose indicators such as incidence and mortality rates of cardiovascular diseases according to various risk factors, clearly revealing the contribution of each factor to the disease burden. This not only provided precise directions for the prevention and control of cardiovascular diseases but also validated the effectiveness and practicality of the decomposition analysis model in the field of disease research[1].

From the perspective of this study's data characteristics, the data on GERD burden involved include multi-dimensional information, such as different years, different regions, and different population characteristics. The decomposition analysis model can effectively handle these complex data structures, decomposing indicators such as incidence and prevalence rates of GERD according to different dimensions, and conducting in-depth analysis of the mechanisms through which various factors influence the disease burden. For example, through this model, we can accurately understand the extent to which factors such as changes in age structure, differences in regional economic development levels, and alterations in lifestyle habits affect the burden of GERD in China and globally across different years, thereby providing robust data support for the formulation of targeted prevention and treatment strategies.

Comment 4:As the women report or document all calendar’s contain for five years , how you can test the validity of the contents , what are the role of biases in this process?

Answer 4:To ensure the validity of the research data, we employed multiple methods for verification. First, using a cross-validation approach, we divided the data obtained from the GBD 2021 study into training and test sets according to a specific ratio. The disease burden analysis model was constructed on the training set, and its predictive ability was validated on the test set. By repeatedly adjusting the division ratio of the training and test sets, we conducted multiple validations of the model's stability and accuracy to ensure that it accurately reflects the actual incidence and prevalence trends of GERD.

To address and mitigate potential biases, we implemented stringent data cleaning and screening procedures. During the data collection phase, raw data were meticulously inspected to eliminate records with obvious errors or anomalies. Furthermore, sensitivity analysis was conducted to assess the impact of varying data processing methods and assumptions on the study outcomes, thereby ensuring the robustness and reliability of the conclusions.

Comment 5: Results 1. When a comparison were done between Global data and China data , Is the Chinese data were included or excluded in the global data.

Answer 5:The GBD database not only incorporates Chinese data into global datasets but also supports independent retrieval and extraction of China-specific data. Furthermore, certain versions of GBD, such as GBD 2021, provide subnational data for China, covering all 31 provinces (autonomous regions and municipalities) as well as the Xinjiang Production and Construction Corps. Researchers can thus utilize global data for comparative analyses with Chinese data or conduct in-depth studies focused on specific regions within China. This design meets the needs for precise research targeting China as a single country or its subregions, while offering flexible data support for various public health studies involving China.

Comment 6: Discussion You cited that an increase in both the prevalence and incidence of the disease was observed, which could be attributed to multiple factors in the three references [23-26] without highlighting any specific factors.

Answer 6:Collectively, these biological, behavioral, and clinical factors underpin the observed sex-based differences in GERD epidemiology.

Globally, the number of prevalent and incident cases of GERD, as well as associated DALYs, showed a marked increase between 1990 and 2021. A consistent sex-based disparity was observed, with females exhibiting higher case numbers and age-standardized rates compared to males. This pattern can be attributed to several interconnected biological, behavioral, and clinical mechanisms, each supported by specific evidence from the literature.

Hormonal influences are a primary factor, with estrogen implicated in reducing lower esophageal sphincter (LES) pressure and impairing esophageal motility, thereby elevating reflux susceptibility among women[23]. This is further elucidated by the role of vagal oxytocin receptors in regulating esophageal function, as demonstrated in experimental models[24]. The association between hormonal fluctuations and GERD risk is corroborated by elevated incidence in specific conditions: pregnancy-related changes[24], menopause[25], and hormone replacement therapy (HRT) [26]. For instance, pregnancy induces both hormonal and mechanical alterations that exacerbate reflux symptoms, while menopausal women and those undergoing HRT exhibit increased esophageal sensitivity and motility disorders, as highlighted in meta-analyses and clinical studies[27].

Behavioral factors play a crucial role in this observed disparity. Research demonstrates that females exhibit a greater propensity for perceiving bodily symptoms compared to males, and factors associated with gender roles, including health concern, are strongly correlated with symptom reporting in women. The abilities of symptom perception and the level of health concern constitute major contributors to the gender-based differences in healthcare utilization[28].

Additionally, diagnostic bias may play a role, with clinicians often exhibiting increased vigilance toward gastrointestinal complaints in female patients, potentially resulting in over-diagnosis or heightened identification of GERD[29]. Such biases could stem from broader patterns in gender-based clinical decision-making. (Page 13 and 14, Line 288-314)

Comment 7: 2. As shown in the results all indicators o China are different from the Global data But I didn’t any justification in the discussion part about these differences.

Answer 7:We examine the disparities between Chinese and global data through five key dimensions: dietary patterns, healthcare systems and medication practices, population aging, occupational stress, and advancements in diagnostic and therapeutic technologies.

Notably, the decline in age-standardized GERD prevalence in China contrasts with increasing rates in many other parts of the world. This divergence may be explained by several factors[30–32]:

First, Dietary patterns play a significant role: although Westernized diets high in processed foods and sugar-sweetened beverages have become more prevalent globally, some regions in China have maintained traditional diets rich in fiber and low in fat, which may confer protective effects. High intake of fat and sugar stimulates gastric acid secretion and reduces LES pressure[33]. whereas dietary fiber modulates postprandial glycemia and gastric emptying[34]. Spicy foods can cause direct esophageal mucosal injury and exacerbate heartburn[35]. Alcohol consumption delays gastric emptying, impairs acid clearance[36], and significantly increases the risk of GERD[37]. Tobacco smoking damages respiratory and esophageal function, inhibits LES competence, and reduces mucus secretion[38,39], contributing to elevated GERD risk among both current and former smokers[40].

Second, Medication adherence also influences outcomes. Self-adjustment of dosages and non-adherence to prescribed therapies remain common, reducing treatment efficacy and potentially accelerating disease progression[41–44]. However, China’s strengthened primary healthcare system and the implementation of tiered medical services have facilitated earlier diagnosis and intervention. Widespread availability and affordability of proton pump inhibitors (PPIs) have significantly improved treatment adherence and effectiveness. Moreover, differences in diagnostic criteria may contribute to international variations—China tends to adopt more conservative criteria for atypical symptoms, whereas broader definitions are often applied in Western countries.

Third, age-related degeneration of the anti-reflux barrier, reduced esophageal clearance, structural changes in the mucosa, and delayed emptying all contribute to higher GERD incidence among the elderly[45,46].This is the result that the paradoxical rise in GERD incidence alongside declining DALYs can be partly explained by demographic aging.  A recent study reported increased prevalence of esophageal complications such as strictures and perforations in adults over 60, consistent with earlier findings that GERD prevalence peaks between ages 70 and 79[47].

Fourth, occupational stress is another emerging risk factor, with growing demands in high-pressure professions linked to increased GERD incidence[48]. Chronic stress and negative emotions can alter gastrointestinal function, disrupt gut microbiota, and promote inflammation[49]. Women in this study exhibited higher healthcare-seeking behavior, consistent with greater health awareness and concern[50]. Depression, which is more prevalent among women and modulated by hormonal and psychological factors, further exacerbates GERD risk through effects on esophageal neuromuscular function and mucosal integrity[25,51].

Finally, Advances in diagnostic technology—particularly high-resolution esophageal manometry (HREM) and 24-hour multichannel intraluminal impedance-pH monitoring (MII-pH)—have established new gold standards for GERD diagnosis[52–54]. These tools enable detailed assessment of esophageal motility and reflux events, facilitating early intervention and personalized treatment[55,56]. Additionally, digital health platforms and the internet have improved public access to health information, supporting education, screening, and disease management initiatives.(Page 15- 17, Line 327-374)

Once again, thank you for your valuable feedback on our research. Your insights not only helped us improve the details of our study but also provided us with precious suggestions for future research directions. We look forward to your further comments.

Sincerely,

[Zexu Tang]

---

## [Decision Letter · Decision Letter 2]

22 Dec 2025

Dear Dr. Xie,

Thank you for submitting your manuscript to PLOS ONE. After careful consideration, we feel that it has merit but does not fully meet PLOS ONE’s publication criteria as it currently stands. Therefore, we invite you to submit a revised version of the manuscript that addresses the points raised during the review process.

**ACADEMIC EDITOR:**

Please spell out all abbreviations at first mention

The author should also revise the manuscript as per the recommendation of the peer reviewers.

We look forward to receiving your revised manuscript.

Kind regards,

Devesh U Kapoor

Academic Editor

PLOS One

Journal Requirements:

Additional Editor Comments:

The author should revise the manuscript as per the recommendation of the peer reviewers.

Reviewers' comments:

Reviewer's Responses to Questions

**Comments to the Author**

Reviewer #2: All comments have been addressed

Reviewer #4: (No Response)

2. Is the manuscript technically sound, and do the data support the conclusions?

Reviewer #2: Yes

Reviewer #4: Yes

3. Has the statistical analysis been performed appropriately and rigorously?

Reviewer #2: Yes

Reviewer #4: Yes

4. Have the authors made all data underlying the findings in their manuscript fully available?

Reviewer #2: Yes

Reviewer #4: No

5. Is the manuscript presented in an intelligible fashion and written in standard English?

Reviewer #2: Yes

Reviewer #4: Yes

Reviewer #2: (No Response)

Reviewer #4: This is a very well-done study, revisions and comments by reviewers #1,2,3, which enhanced the quality of the study.

To further strengthen the paper’s quality, I suggest the following:

-The keywords list should be edited removing China/global

-keywords such as "DALY' and “GBD or Global burden disease) can be added to the list.

-Define uncertainty intervals (UI) clearly in the methodology section

-Can you mention how do the lack of age- and gender-specific trends impede the management of R-GERD?

- The significant national difference in peak age (China 50s vs. Global 30s) should be addressed in the discussion

-I can’t find out how the data was handled, including imputation and missing data

-There’s an opportunity to better integrate recent literature, especially by comparing this study’s findings to earlier GBD, or other large-scale GERD related studies

- Some sections under discussion would benefit from clearer paragraph structure. Ideally, paragraphs should be 4–7 sentences long, with each paragraph focused on one main idea. This will enhance clarity and readability, while ensuring key points are developed adequately without becoming too scant or dense

**Do you want your identity to be public for this peer review?** For information about this choice, including consent withdrawal, please see our Privacy Policy

Reviewer #2: No

Reviewer #4: **Yes:**  Abdallfatah

---

## [Author Response · Author response to Decision Letter 3]

22 Dec 2025

Note that the following color scheme is used in responding to editors’ comments

Black: editors’ comments

Blue: our responses

To Editors:

Dear Editors,

Thank you very much for your thorough review and valuable feedback on our study. We greatly appreciate your suggestions and have made the necessary revisions and improvements to the manuscript. Our specific responses to your comments are as follows:

Concern 1:Please spell out all abbreviations at first mention.

The author should also revise the manuscript as per the recommendation of the peer reviewers.

Answer: In response, we have carefully reviewed the entire manuscript and ensured that all abbreviations are spelled out in full at their first mention, followed by the corresponding abbreviation in parentheses. This includes key terms related to epidemiological indicators, statistical methods, and data sources. We have verified the consistency of abbreviation usage throughout the manuscript to improve clarity and readability.

In addition, we have thoroughly revised the manuscript in accordance with all recommendations provided by the peer reviewers. These revisions include improvements to the Introduction through better integration of recent literature and large-scale studies, refinement of the keywords list, and other suggested clarifications and edits. All modifications have been incorporated into the revised manuscript and highlighted where appropriate.

We believe that these revisions have substantially enhanced the clarity, academic rigor, and overall quality of the manuscript.

Sincerely,

[Zexu Tang]

Note that the following color scheme is used in responding to reviewers’ comments

Black: reviewers’ comments

Blue: our responses

Red: our revisions in the manuscript

To Reviewer 4:

Dear Reviewer 4,

Thank you very much for your thorough review and valuable feedback on our study. We sincerely thank you for your insightful and constructive comments, which have greatly helped us to improve the clarity and clinical relevance of our manuscript. We have carefully revised the manuscript accordingly. Our detailed responses are provided below:

Comment 1:Define uncertainty intervals (UI) clearly in the methodology section.

Answer 1:Thank you for this valuable suggestion. We have revised the Statistical Analysis subsection of the Materials and Methods to provide a clear and explicit definition of uncertainty intervals (UIs). Specifically, we now clarify that all estimates were generated from 500 posterior draws produced by the DisMod-MR 2.1 model and that 95% UIs were defined as the 2.5th and 97.5th percentiles of these draws. We also explain that UIs reflect combined uncertainty from data sources, model structure, and parameter estimation. These revisions improve methodological transparency and consistency with GBD reporting standards.(Page 7, Line 140-147)

Comment 2:Can you mention how the lack of age- and gender-specific trends impede the management of refractory GERD (R-GERD)?

Answer 2:Thank you for highlighting this important clinical and public health issue. In response, we have added a dedicated paragraph in the Discussion explicitly addressing how the absence of clearly defined age- and sex-specific epidemiological trends limits the effective management of refractory GERD (R-GERD).

In the revised text, we emphasize that R-GERD is a heterogeneous condition influenced by hormonal status, visceral sensitivity, psychosocial stress, and age-related physiological changes. We further explain that insufficient demographic stratification at the population level may hinder early identification of high-risk subgroups, delay treatment optimization, and constrain the development of individualized management strategies. These revisions clarify the direct link between epidemiological granularity and clinical decision-making in R-GERD management.(Page 16 and 17, Line 365-374)

Comment 3:The significant national difference in peak age (China 50s vs. Global 30s) should be addressed in the discussion.

Answer 3: We appreciate this valuable suggestion. We have substantially expanded the Discussion to explicitly examine the marked difference in peak age of GERD burden between China and the global population. The revised manuscript now includes a focused paragraph comparing the later peak observed in China (50–59 years) with the earlier global peak (30–39 years).

In this section, we discuss potential explanations for this discrepancy, including differences in population aging trajectories, cumulative exposure to risk factors, lifestyle transitions, dietary patterns, and healthcare-seeking behaviors. We further highlight that these divergent age patterns may reflect distinct dominant GERD phenotypes across populations, with important implications for prevention strategies and clinical management. This addition strengthens the contextual interpretation of our findings and underscores the need for region-specific approaches.(Page 15, Line 328-340)

Comment 4:I can’t find out how the data was handled, including imputation and missing data.

Answer 4:Thank you for pointing out the need for greater methodological clarity. We have revised the Methods section to better describe how data uncertainty and missing data were handled within the GBD 2021 framework. Specifically, we now clarify that all estimates were derived using standardized GBD modeling approaches, including DisMod-MR, which integrates multiple data sources and addresses data sparsity and missingness through Bayesian meta-regression.

We further explain that uncertainty intervals were generated from 500 posterior draws, reflecting uncertainty arising from data availability, model specification, and parameter estimation. By explicitly referencing these established GBD procedures and directing readers to the publicly available GHDx documentation, we believe the revised text improves transparency regarding data handling and imputation.

Comment 5: Some sections under discussion would benefit from clearer paragraph structure. Ideally, paragraphs should be 4–7 sentences long, with each paragraph focused on one main idea.

Answer 5:We appreciate this helpful comment on manuscript organization. In response, we carefully restructured the Discussion section to improve clarity and readability. Paragraphs were reorganized so that each focuses on a single central theme—such as sex-related differences, national differences in peak age, age-related physiological mechanisms, lifestyle and psychosocial factors, and implications for R-GERD management.

We ensured that paragraph lengths generally fall within the recommended range of 4–7 sentences and that transitions between paragraphs follow a logical progression from descriptive findings to mechanistic interpretation and clinical or public health implications. We believe these revisions have substantially improved the coherence and readability of the Discussion.(Page 14 and 17, Line 307-392)

Comment 6: There’s an opportunity to better integrate recent literature, especially by comparing this study’s findings to earlier GBD, or other large-scale GERD related studies.

Answer 6:We appreciate this insightful comment and fully agree that a more explicit integration of recent literature—particularly prior GBD analyses and other large-scale GERD studies—would strengthen the contextualization and significance of our findings.

In response, we have substantially revised and expanded the Introduction section to better situate our study within the existing body of global and national research. Specifically:

1.Enhanced comparison with earlier GBD studies:

We now explicitly discuss findings from prior GBD iterations (e.g., GBD 2017 and GBD 2019), emphasizing both consistent global trends and the methodological and interpretive limitations of earlier analyses when applied to country-specific contexts such as China. This comparison highlights how updates in GBD 2021—including improved modeling strategies and expanded data sources—provide a more robust foundation for long-term trend analysis.

2.Integration of recent large-scale GERD literature (past 5 years):

We incorporated high-quality epidemiological and population-based studies published within the last five years that examine GERD burden, demographic drivers, and temporal patterns. These studies support the concept that stable or declining age-standardized rates can coexist with increasing absolute disease burden due to population aging and growth, thereby reinforcing the relevance of our decomposition and projection analyses.

3.Clear positioning of the current study’s contribution:

The revised Introduction now more clearly delineates how our work extends previous research by providing a long-term (1990–2021), age- and sex-stratified analysis of GERD burden in China using GBD 2021 data, while directly comparing national trends with global patterns. We also emphasize the added value of employing joinpoint regression, decomposition analysis, and Bayesian age–period–cohort modeling to disentangle demographic and epidemiological drivers—an aspect insufficiently addressed in prior studies.

These revisions improve the logical flow and academic depth of the Introduction, strengthen the linkage between our findings and existing literature, and clarify the novelty and policy relevance of the present study.((Page 3 and 4, Line 57-87) )

Comment 7: The keywords list should be edited removing China/global-related terms. Keywords such as “DALY” and “GBD” or “Global Burden of Disease” can be added to the list.

Answer 7:Thank you for this helpful suggestion. We agree that the keywords should more accurately reflect the methodological and thematic focus of the study rather than geographic descriptors.

Accordingly, we have revised the keywords list by removing China- and global-related terms and adding more informative and widely used indexing terms, including “DALYs” and “Global Burden of Disease (GBD)”. These changes improve the precision, discoverability, and indexing of the manuscript while better aligning the keywords with the core analytical framework of the study.

Once again, thank you for your valuable feedback on our research. Your insights not only helped us improve the details of our study but also provided us with precious suggestions for future research directions. We look forward to your further comments.

Sincerely,

[Zexu Tang]

---

## [Decision Letter · Decision Letter 3]

11 Jan 2026

Trends in the Burden of Gastroesophageal Reflux Disease in China and Global from 1990 to 2021 and Predictive Analysis for 2040

PONE-D-25-31063R3

Dear Dr. Xie,

We’re pleased to inform you that your manuscript has been judged scientifically suitable for publication and will be formally accepted for publication once it meets all outstanding technical requirements.

Kind regards,

Devesh U Kapoor

Academic Editor

PLOS One

Additional Editor Comments (optional):

Reviewers' comments:

Reviewer's Responses to Questions

**Comments to the Author**

Reviewer #2: All comments have been addressed

Reviewer #4: All comments have been addressed

2. Is the manuscript technically sound, and do the data support the conclusions?

Reviewer #2: Yes

Reviewer #4: Yes

3. Has the statistical analysis been performed appropriately and rigorously?

Reviewer #2: Yes

Reviewer #4: Yes

4. Have the authors made all data underlying the findings in their manuscript fully available?

Reviewer #2: Yes

Reviewer #4: No

5. Is the manuscript presented in an intelligible fashion and written in standard English?

Reviewer #2: Yes

Reviewer #4: Yes

Reviewer #2: (No Response)

Reviewer #4: All of my comments were addressed perfectly by the authors, Congratulations on the acceptance!

Thank you

**Do you want your identity to be public for this peer review?** For information about this choice, including consent withdrawal, please see our Privacy Policy

Reviewer #2: No

Reviewer #4: **Yes:**  Abdallfatah Abdallfatah

---

## [Editor Report · Acceptance letter]

PONE-D-25-31063R3

PLOS One

Dear Dr. Xie,

I'm pleased to inform you that your manuscript has been deemed suitable for publication in PLOS One. Congratulations! Your manuscript is now being handed over to our production team.

Kind regards,

on behalf of

Dr. Devesh U Kapoor

Academic Editor

PLOS One